# Modeling the interplay of sex hormones in cardiac hypertrophic signaling

Adhithi Lakshmikanthan[1,2], Minnie Kay[1,2], Pim J.A. Oomen[1,2,3]*

**1** Department of Biomedical Engineering, University of California, Irvine, Irvine, California, United States of America, **2** Edwards Lifesciences Foundation Cardiovascular Innovation and Research Center, University of California, Irvine, Irvine, California, United States of America, **3** Center for Complex Biological Systems, University of California, Irvine, Irvine, California, United States of America

* pim.oomen@uci.edu

## Abstract

Biological sex plays a crucial role in the outcomes of cardiac health and therapies. Sex hormones are known to strongly influence cardiac remodeling through intracellular signaling pathways, yet their underlying mechanisms remain unclear. To address this need, we developed and validated a logic-based systems biology model of cardiomyocyte hypertrophy that, for the first time, incorporates the effects of both estradiol (E2) and testosterone (T) alongside well-established hypertrophic stimuli (Strain, angiotensin II (AngII), and endothelin-1 (ET-1)). We qualitatively validated the model to literature data with 82% agreement. Quantitative validation was done by simulating the impact of the inputs (E2, T, Strain, AngII, and ET-1) on cardiac hypertrophy, captured as change in CellArea. We perturbed the validated model to examine the differential response to hypertrophy and identify changes in influential and sensitive downstream nodes for a pre-menopausal female, post-menopausal female, younger male, and older male condition. Our model shows how T and E2 interact with each other and other hypertrophic stimuli, with T demonstrating a more potent hypertrophic effect than E2. This model increases our understanding of the mechanisms through which sex hormones influence cardiac hypertrophy and can aid with developing more effective cardiac therapies for all patients.

## Author summary

Differences between female and male hearts extend far beyond size and structure. Sex hormones estradiol and testosterone play key roles in sex-specific cardiac remodeling via intracellular pathways. Understanding how these sex hormones impact cardiac remodeling is critical for developing more effective, sex-specific approaches to cardiovascular care. Logic-based systems biology models have proven useful in quantifying and analyzing complex and intricate intracellular signaling network dynamics in various cell types. We leverage this method to

**Data availability statement:** All model code and data used to generate the results in this study are available on our lab's GitHub page (http://www.github.com/BEATLabUCI). This repository includes: Python code developed to run and analyze the model; annotated Jupyter notebooks that provide guidance on using the code and reproducing the results and figures; Excel file containing all species and reactions, and their parameters and sources.

**Funding:** This work was supported by grants from the National Institutes of Health (NIH) grant R01HL159945 and American Heart Association Career Development Award 10.58275/AHA. 25CDA1452893.pc.gr.229647, awarded to PJAO, which also provided partial support for AL. The funders had no role in study design, data collection and analysis, decision to publish, or preparation of the manuscript.

**Competing interests:** The authors have declared that no competing interests exist.

develop a model of cardiomyocyte hypertrophy, which, for the first time, includes the effect of both estradiol and testosterone. Considering the combined influence of these hormones is important because these hormones vary in both women and men throughout their lives. The model was developed and validated based on previously published experimental studies. We then investigated differences in cardiomyocyte hypertrophy in pre- and post-menopausal women, and, younger and older men.

## Introduction

In the past decade it has become increasingly clear that biological sex plays a crucial role in the outcomes of cardiac health and therapies. The female heart is known to be smaller than the male heart, yet their differences extend well beyond size. Indeed, sex differences exist across the scales, from the organ itself down to the individual cells [1]. These differences have been attributed to sex hormones, sex chromosomes, epigenetics, and autosomal gene regulation variations [2–4]. The sex hormones estradiol (E2) and testosterone (T) in particular influence more than 70% of sex-specific differences in cardiac gene expression [4]. Therefore, understanding these hormonal influences is critical for developing more effective, sex-specific approaches to cardiovascular care.

E2 and T are known to strongly affect cardiomyocyte hypertrophy through intracellular signaling pathways [1,5,6]. The impact of these hormones on hypertrophy is most evident during two key hormonal transition periods: (1) puberty and (2) menopause. First, at birth, female hearts are slightly larger than male hearts, but this difference reverses at puberty [7–9], coinciding with increased pro-hypertrophic T in males and anti-hypertrophic E2 in females [1,5,6]. While the number of cardiomyocytes, which make up ~80% of myocardial volume, remains mostly unchanged across sex during puberty, male cardiomyocytes are larger, underscoring the hormonal influence on cardiac growth through hypertrophy [10]. Second, premenopausal women, with a higher E2:T ratio, have a lower risk of hypertrophic heart failure compared to men, who have a lower E2:T ratio. However, this anti-hypertrophic advantage diminishes after menopause as the E2:T ratio becomes becomes progressively similar between the sexes [11]. Thus, the E2 and T cross-talk is different between the sexes and changes significantly with age, as does its effect on cardiac remodeling through hypertrophy [12,13].

While sex hormones are now known to be crucial in sex-specific cardiac remodeling, the mechanisms driving these differences remain poorly understood. Due to the complexity and intricate interplay of processes involved in cardiac remodeling, computational models have proven useful in quantifying and analyzing these dynamics. Logic-based systems biology models, in particular, have been used to explore molecular mechanisms in cell-level remodeling. Originally developed by Kraeutler et al. [14], these models have since been used to investigate signaling pathways in various cells and tissues including fibroblasts [15,16], cardiomyocytes [17,18], skeletal muscle [19], and epidermis [20]. More recently, they were also used to study the

influence of E2 on drug response in cardiac fibroblasts [21] and cardiomyocyte hypertrophy during and after pregnancy [22]. Note that the role of T was never considered in any of these models. Logic-based systems biology models are developed by creating a network structure derived from individual, often qualitative, known relationships from the literature. These relationships include observations such as a 'kinase X' up-regulating phosphorylation of a 'protein Y'. The relationships are modeled by normalized Hill-type ordinary differential equations to capture cooperative binding mechanisms and saturation effects even in the absence of actual kinetic data [23,24]. Thus, logic-based network models capture individual phosphorylation and gene transcription crosstalk as a system of ordinary differential equations. These models overcome the parameterization issues that are often encountered in complex, detailed intracellular signaling models by using a small set of parameters and a discrete set of rules to update the state of each variable.

In this study, we develop and validate a logic-based systems biology model to provide mechanistic insight on the impact of sex hormones (E2 and T) on cardiomyocyte hypertrophy. We consider sex hormonal influence and their crosstalk with well-established hypertrophic stimuli - strain, angiotensin II (AngII) and endothelin-1 (ET-1). The network architecture is constructed using 27 individual in vivo and in vitro literature studies. The reactions of this network are modeled using normalized Hill-type ordinary differential equations. We qualitatively validate our model predictions against individual perturbations, as well as their modulations with E2 or T reported in the literature and observe an 82% agreement. To quantitatively validate the model's hypertrophic response to single and combined input perturbations, we simulate three independent experimental (in vitro and in vivo) sex-hormone studies. Finally, we analyze the model's behavior under premenopausal female, post-menopausal female, younger male and older male concentrations of E2 and T to identify the most influential and sensitive cardiomyocyte hypertrophic nodes across varying levels of these hormones. Interestingly, the results suggest that T has a greater influence on cardiomyocyte hypertrophy than E2 in our model. Altogether, our model provides a much-needed tool to gain mechanistic insights in sex-hormone driven cardiac hypertrophy.

## Results

### Network structure

It is well established that cardiac hypertrophy is heavily influenced by hormones such as AngII and ET-1, and mechanical Strain. However, the impact of sex hormones E2 and T and their cross-talk with these established hypertrophic stimuli has not been extensively researched. To address this gap, we developed a systems biology network model with Strain, AngII, ET-1, as well as E2 and T as model inputs, and a change in cardiomyocyte area (CellArea) as the model output. In vivo and in vitro studies on female and male specimens were used to develop a sex hormone-driven signaling model of cardiomyocyte hypertrophy (Fig 1). Where possible, we focused on rodent studies. The network structure was developed based on individual signaling pathways identified through western blot analyses from 27 literature studies. The resulting network comprises 29 unique nodes, representing various biomolecules (including proteins, kinases, and transcription factors), generic input (Strain), generic composite output (CellArea) with 49 edges, indicating inhibition or activation reactions. AND and OR gates were used to model interactions involving multiple species. The network reactions were modeled using normalized Hill-type ordinary differential equations (ODEs) [14]. The system of ODEs was generated using Netflux [25]; a more detailed description of its mathematical formulation can be found in the Methods section.

In the resulting network, E2 and T are connected to a similar amount of downstream nodes, 27 and 24, respectively. We additionally assessed bottlenecks in our network, defined as how often a node lies on the shortest path thus making it influential. The top 3 bottlenecks in our network were ERK12, eNOS and NFAT. E2 attenuates Strain-driven activation of p38 mitogen-activated kinases (p38), extracellular signal-regulated kinases 1/2 (ERK1/2), and attenuates Strain inhibition of phosphoinositide 3-kinase (PI3K)/protein kinase B (Akt). E2 attenuates AngII-driven activation of ERK1/2 while T amplifies it. Further, T also supplements AngII activation of calcineurin (CaN). E2 attenuates ET-1 driven activation of ERK1/2, and attenuates ET-1 inhibition of PI3K. We opted for a single composite output, CellArea, to represent cell hypertrophy, rather than other individual protein outputs like ANP and BNP. This decision was made to focus specifically on absolute

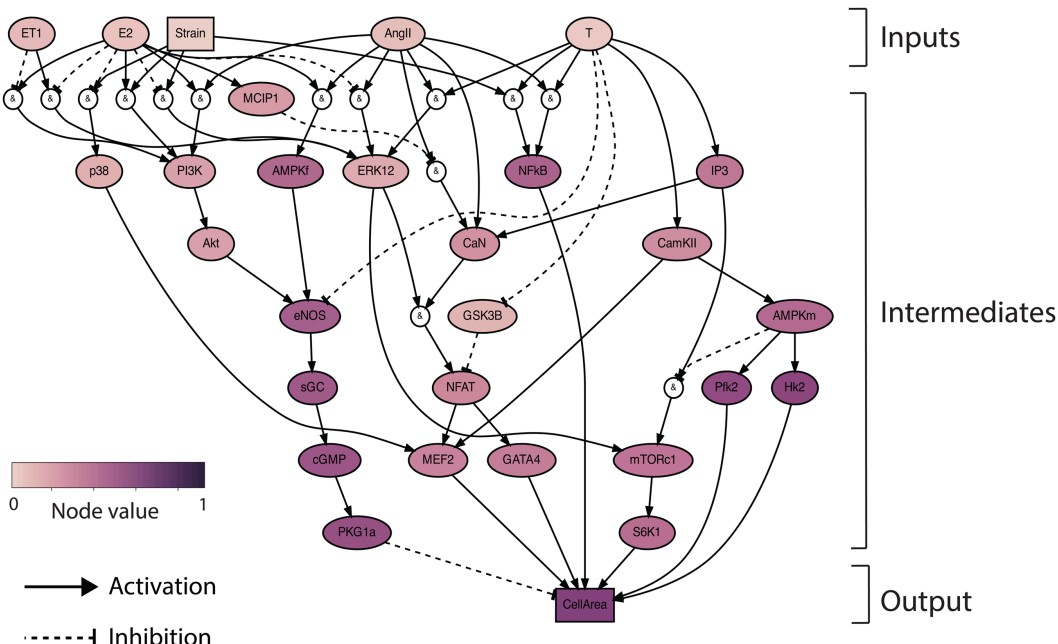

**Fig 1**. **Network model schematic: The sex-hormone driven cardiomyocyte hypertrophy model was built using 27 experimental studies with E2, T, Strain, AngII, ET-1 as inputs and change in CellArea as output.** The model network consists of 29 unique nodes and 49 edges, with AND and OR gates to model multi-species interactions. AND gates are shown by '&' and OR gates exist when more than 1 node influences a downstream node.

changes in cardiomyocyte size. Strain and stress are constitutively related and we expect both to increase during aortic banding, however we refer to mechanical change due to pressure overload as a composite Strain input in our model. In this network, out of the 47 reactions, 9 came from female specimens, 13 from males, and 25 from unspecified sex. There were 12 in vivo papers, 22 in vitro papers, and 13 included both in vivo and in vitro studies [26–51].

Sex hormone-driven reactions in the model were explicitly informed by hormone measurements in the literature studies. One exception to this is the reaction based on a male androgen receptor (AR) knockdown animal model that was used to demonstrate the effects of T and AngII on ERK1/2, where the influence of T was inferred rather than directly measured [31]. The second exception is where the influence of T on eNOS was inferred, as the E2/eNOS activation is seen in females, but not in males [42]. This sex difference is hypothesized to be due to a couple of potential reasons such as VEGF's role in males, differences in Cav-3 protein which plays a big role in eNOS activation, or, the number of ERβ between the sexes [42]. Additionally, the decision to split AMPK into AMPKm and AMPKf is because we had experimental data from specific sexes, such as Hk2 and Pfk2 being downstream of AMPK in males, and, E2 and AngII acting on AMPK in females. This separation allows for a more nuanced expression of the sex-specific effects on AMPK signaling in the model.

## Qualitative validation

We set uniform Hill parameters ($EC_{50} = 0.5$ and $n = 1.4$) across all reactions, following Ryall et al. [17]. We also calibrated our model by setting uniform input node weights that resulted in normalized (CellArea = 0.5), allowing for equal amounts of hypertrophy (CellArea >0.5) and atrophy (CellArea <0.5). We then qualitatively validated our model by following established logic-based modeling approaches, where the model outputs in response to input up-regulations are compared to literature data. We represent up-regulation of each node by increasing their value by $\Delta y = 0.3$, based on Irons et al. [52]. The quality of fit of the response was determined by comparing the decrease ($\Delta y < -0.05$), increase ($\Delta y > 0.05$), or no

change ($-0.05 \leq \Delta y \leq -0.05$) of downstream node activity with literature data. Our model's response to the perturbations matched 82% of the literature-reported data, shown in Fig 2.

We analyzed model responses to individual stimuli (E2, T, Strain, AngII, and ET-1) relative to baseline, and their combined effects with sex hormones (+E2 or +T) relative to each stimulus (Strain, AngII, ET-1) alone. The literature studies used to build the model found no change in ERK12 and CellArea due to E2 by itself, but the model predicted a slight decrease. Similarly, we predicted an increase in PI3K due to E2 when literature reported no change. The model showed no change in PI3K, while the data suggests that ET-1 should decrease PI3K, and, AngII and Strain should increase PI3K. Finally, the model did not capture AngII's activation of AMPKf. E2 mitigated the effects of the pro-hypertrophic stimuli Strain, AngII and ET-1, as seen in AngII+E2, ET-1+E2 and Strain+E2 cases, while T exacerbated pro-hypertrophic effects of Strain and AngII. Besides CellArea, the model was also able to capture more nuanced changes in the intermediary nodes. For example, an increase of only AngII had no effect on MCIP1, while increasing AngII and E2 induced an increase in MCIP1. We ensured that a steady-state was established in the baseline simulation and in the single perturbation simulation before applying further perturbations.

## Quantitative validation

Logic-based network models offer a unique advantage: they can be developed using *qualitative* data, and then provide *quantitative* predictions. We validated our model by testing its ability to predict how biochemical stimuli affect rat cardiomyocyte hypertrophy using independent studies, i.e. studies not used in our model's development. We tested the model's ability to capture E2's anti-hypertrophic effects by comparing our predictions to in vitro experiments from Pedram et al. [53], where they measured changes in cardiomyocyte cell size in responses to AngII and ET-1 with and without E2 in

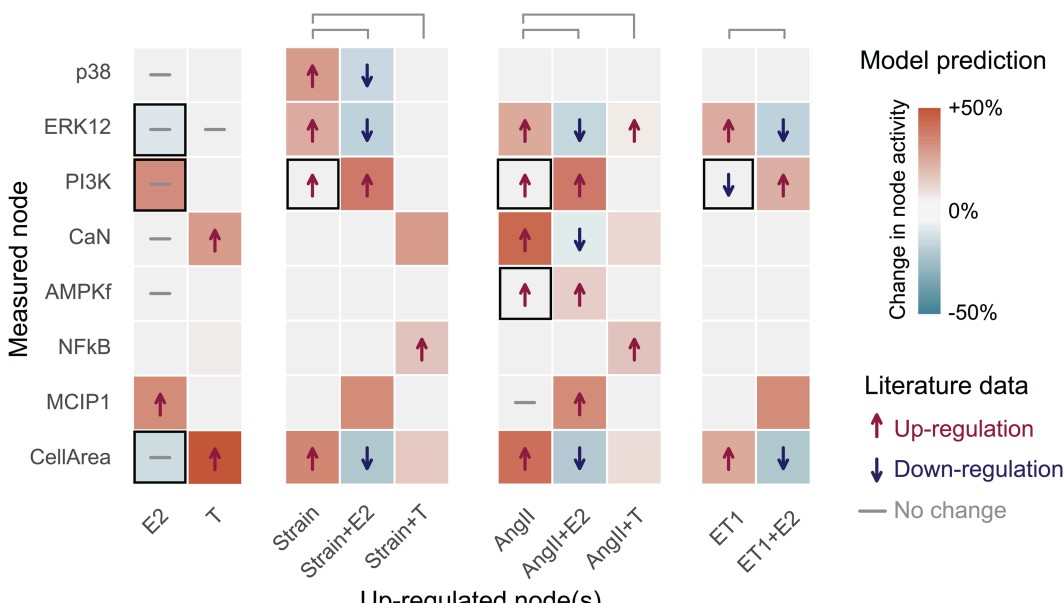

**Fig 2**. **Qualitative validation: 32/39 (82%) of the model nodes were correctly validated against literature data.** Literature data is shown with arrows for up- or down-regulation or a dash for no change. If the effect of an up-regulation was not found in the literature, there are no symbols. The model predictions are shown with a change in background color–red indicates increased and blue decrease activity, with shades of color representing the different intensities, and white represents no change. Model responses to individual stimuli (E2, T, Strain, AngII and ET-1) were calculated relative to baseline, while their combined effects with sex hormones (+E2 or +T) were calculated relative to each stimulus alone, e.g. the model response to 'Strain+E2' was compared to Strain. Each set of perturbations are separated by white space between them.

pooled neonatal and pregnant rat cardiomyocytes (Fig 3a). We correctly matched increases in CellArea due to AngII and ET-1. We overestimated AngII mitigation by E2 (95.7% predicted vs. reported 38±5% change in CellArea), and overestimated E2's mitigation with ET1 (48% predicted vs. reported 33±4.5% change in CellArea). For the in vivo studies, we overestimated the impact of strain: our model predicted a 152% change of CellArea compared to the observed 41.7±3.25 % change induced by aortic banding in female rats by Donaldson et al. [54], and 32.7% vs. 21±2.15% for its mitigation by E2 (Fig 3b). For a spontaneous hypertensive rat model from Chen et al. [55], we underestimated the anti-hypertrophic effect of E2 (–5% predicted vs. reported –14±1% change in CellArea) and underestimated the combined effect of E2 and T (–2% predicted vs. reported 19±1.25% change in CellArea). We ensured that a steady-state was established in the baseline simulation and in the single perturbation simulation before applying further perturbations. We set the model input weights to represent the experimental conditions, see Methods section for more details.

## Cross-talk between hypertrophic stimuli

The cardiomyocyte hypertrophy model correctly predicts that E2 mitigates hypertrophy while the other inputs (T, Strain, AngII and ET-1) promote hypertrophy. A 2-D landscape plot (Fig 4) shows the effect of input interactions on CellArea. While these individual relationships between input nodes and changes in cell area have been documented in the literature, our model is capable of perturbing the effects of interactions between input species on hypertrophy. The model predicts that an increase in E2 can overcome the hypertrophic effect of Strain, AngII, ET-1, and T, allowing for higher concentrations of these nodes before hypertrophy occurs. This effect is seen for Strain, AngII, and ET-1, with E2 > 0.7 preventing hypertrophy. The model also predicts that relatively small increase of T concentration > 0.3 exacerbates the hypertrophic effect of Strain, AngII and ET-1. Furthermore, Strain, AngII, and ET-1 exacerbate each others' pro-hypertrophic effects.

A global sensitivity analysis using Sobol's method (Fig 5 a,b) showed T and E2 to have a higher influence on many downstream nodes when compared to other inputs, with T being more influential on CellArea than E2. We also see which nodes are influenced to a greater extent by each of the inputs, which will be important in the future when understanding the influence of drugs on specific pathways.

## Case studies: comparisons between sex and age

We use the sex-hormone network model to predict how varying sex hormone concentrations in pre-menopausal females, post-menopausal females, younger and older males affects the response of downstream nodes and, consequently, cardiomyocyte response to hypertrophy. We calibrated E2 and T to create pre-menopausal female, post-menopausal female, younger and older male models (see Methods for more details). Since Strain, AngII, and ET-1 inputs do not vary between

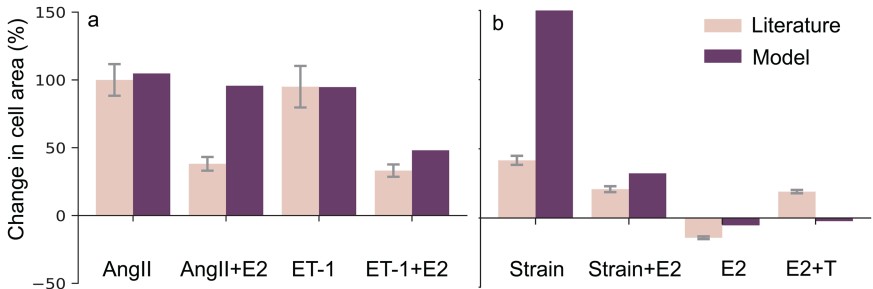

**Fig 3**. Quantitative validation: Model is validated (a) against in vitro literature data from Pedram et al. [53] showing AngII and ET-1 increase CellArea while E2 attenuates that increase, and, (b) against in-vivo literature data from Donaldson et al. [54] and Chen et al. [55] showing Strain and T increasing CellArea, while E2 attenuates that increase. The whiskers indicate the standard deviation of the experimental data. Note that the model predicted a near-zero change in CellArea to the 'Strain+E2' and 'E2+T' conditions.

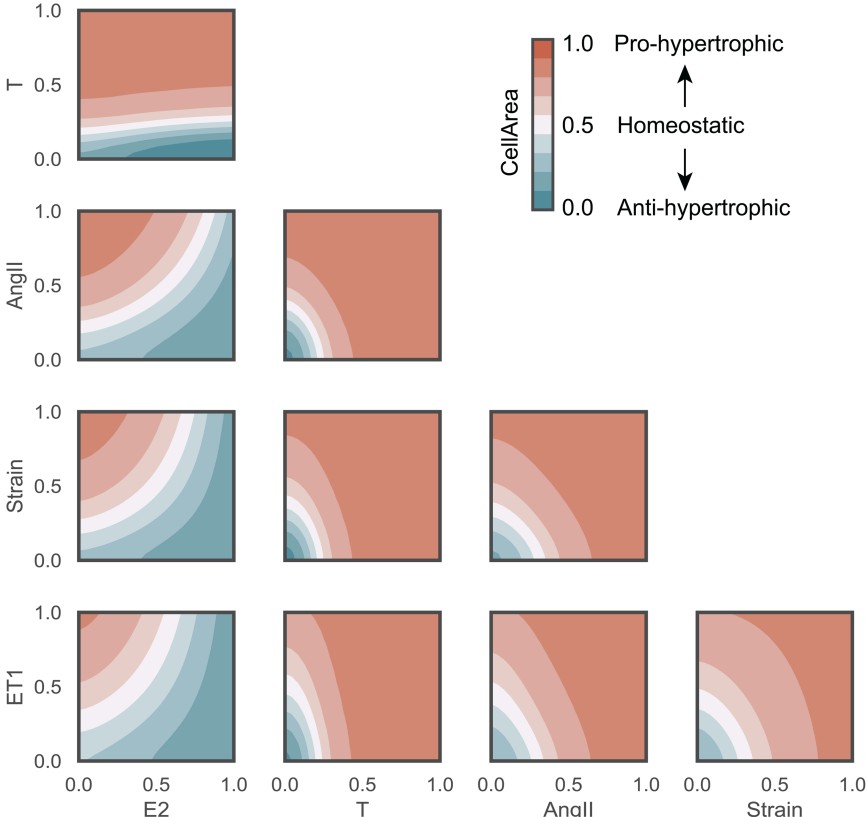

**Fig 4**. **Network model cross-talk: Combined input perturbations predict that E2 mitigates hypertrophic effects of the other stimuli while T exacerbates them.** T, AngII, Strain, and ET-1 amplify each others' hypertrophic effects. Normalized CellArea = 0.5 represents a homeostatic state (white), CellArea > 0.5 pro-hypertrophy (red shades), and CellArea < 0.5 anti-hypertrophy (blue shades).

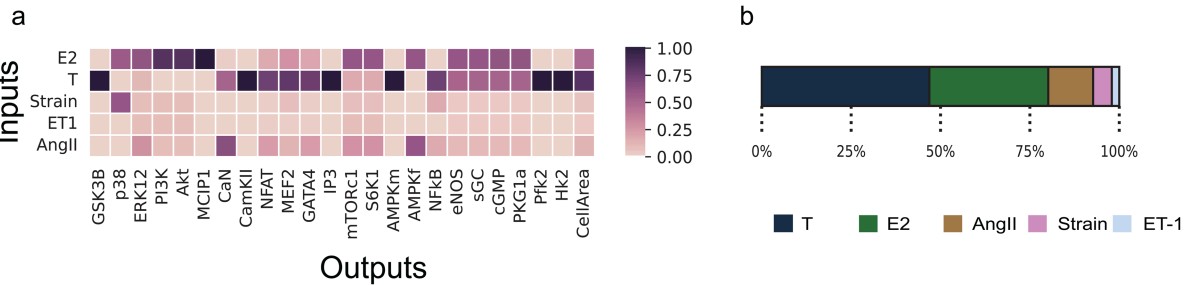

**Fig 5**. **(a) Global sensitivity analysis and (b) Input parameter sensitivity using Sobol's method identifies influential nodes.**

healthy men and women [56,57], we kept these inputs constant. We perturbed the model by applying up-regulation and knock-down of each individual node, as in Watts et al. [21], and identified the most influential and sensitive nodes. Influence is defined as the sum of absolute change in activity of all other nodes upon knockdown/up-regulation of the node, while sensitivity is defined as the sum of absolute change in activity of the node across all knockdown/up-regulation simulations. This perturbation analysis revealed interesting differences in influence and sensitivity between the three models.

During up-regulation, T is found to be more influential than all other nodes in all four models (Fig 6a), though its magnitude is lesser in pre-menopausal females. During knock-down analysis (Fig 6b), endothelial nitric oxide synthase (eNOS) becomes the most influential node for all four conditions. We note a slightly higher influence of cGMP in post-menopausal females and older males during knock-down. CellArea and PKG1 are the most sensitive node during up- and down-regulation for all 4 models during knock-down. While the post-menopausal, younger and older male models have similar influence and sensitivity scores, there are differences in the pre-menopausal model, potentially due to a higher E2/T ratio. The increase in Akt's sensitivity during knock-down in the pre-menopausal condition might be significant for hypertrophic regulation.

Of note are the differences in E2 influence in both the up-regulation and knockdown analysis. In the up-regulation analysis, the influence of E2 is greater in the post-menopausal and male models relative to the pre-menopausal model. Conversely, the influence of E2 in the knockdown analysis is greater in the pre-menopausal model relative to the post-menopausal and male models. This occurs because the baseline pre-menopausal model already has high levels of E2, making its knockdown more impactful than further up-regulation of E2, which is already elevated. The greater influence of E2 in the male and post-menopausal models indicates its potential to mitigate hypertrophy in these groups, who naturally have low E2. In the older male model with 20% reduced T, the influence of E2 during up-regulation analysis decreases when compared to a male with higher T or a post-menopausal female model, as there is lesser hypertrophy present. We also notice the magnitude of CellArea sensitivity in the older male model during upregulation and knock-down analysis to be similar to the post-menopausal females. The post-menopausal, male and older male models behave more similarly to one another.

We also used the four models to analyze the response curves of CellArea as a function of the signaling input nodes (E2, T, Strain, AngII, ET-1) and intermediary nodes (eNOS, glycogen synthase kinase-3 (GSK3$\beta$), calmodulin-dependent protein kinase (CamKII)) that were found to be influential in the up-regulation and knock-down analysis. These response curves can be seen in Fig 7. The E2 and T response curves show their anti- and pro-hypertrophic effects, respectively. We also note a more moderate CellArea response to changes in E2 compared to T, which converges around T = 0.5. The response curves of the other model inputs show interesting differences between the three models. For example, increasing Strain leads to an increased CellArea for all four models, but there are some interesting differences. The younger male model's (high T) curve is shifted upwards compared to the pre-menopausal female model (high E2). Interestingly, the post-menopausal model (low E2) initially starts close to the pre-menopausal curve but eventually (at Strain > 0.5) matches the younger male curve, indicating that this model did not have sufficient E2 to prevent hypertrophy at high Strain levels when compared to the pre-menopausal female model. The older male curve starts lower than the younger male curve due to lower T, but eventually (at Strain > 0.5) matches the younger male and post-menopausal curves. Together, these differences underline the hypertrophic effect of T and the protective effect of E2, which we previously noted in Fig 4. A similar behavior can be seen in the AngII and ET-1 curves. The non-input intermediary nodes eNOS, GSK3$\beta$, and CamKII exhibited high influence in the up-regulation and knock-down analysis. It is interesting to see the post-menopausal female curve closer to males in GSK3$\beta$ and CamKII which further highlights the importance of considering the importance of both sex hormones E2 and T together. The older male model with 20% reduced T is closer to the post-menopausal model than the younger male model for specific nodes at low concentrations (in the case of T, Strain, AngII, ET-1, CamKII), or higher concentrations (in the case of eNOS and GSK3$\beta$).

Altogether, these predictions underscore the model's potential to predict age-like and sex-specific hypertrophic responses to combinations of various stimuli. Recall that E2 and T are connected to a similar number of nodes in the network. Therefore, these responses arise from the interactions and values of the input and intermediary nodes, rather than any structural bias within the network.

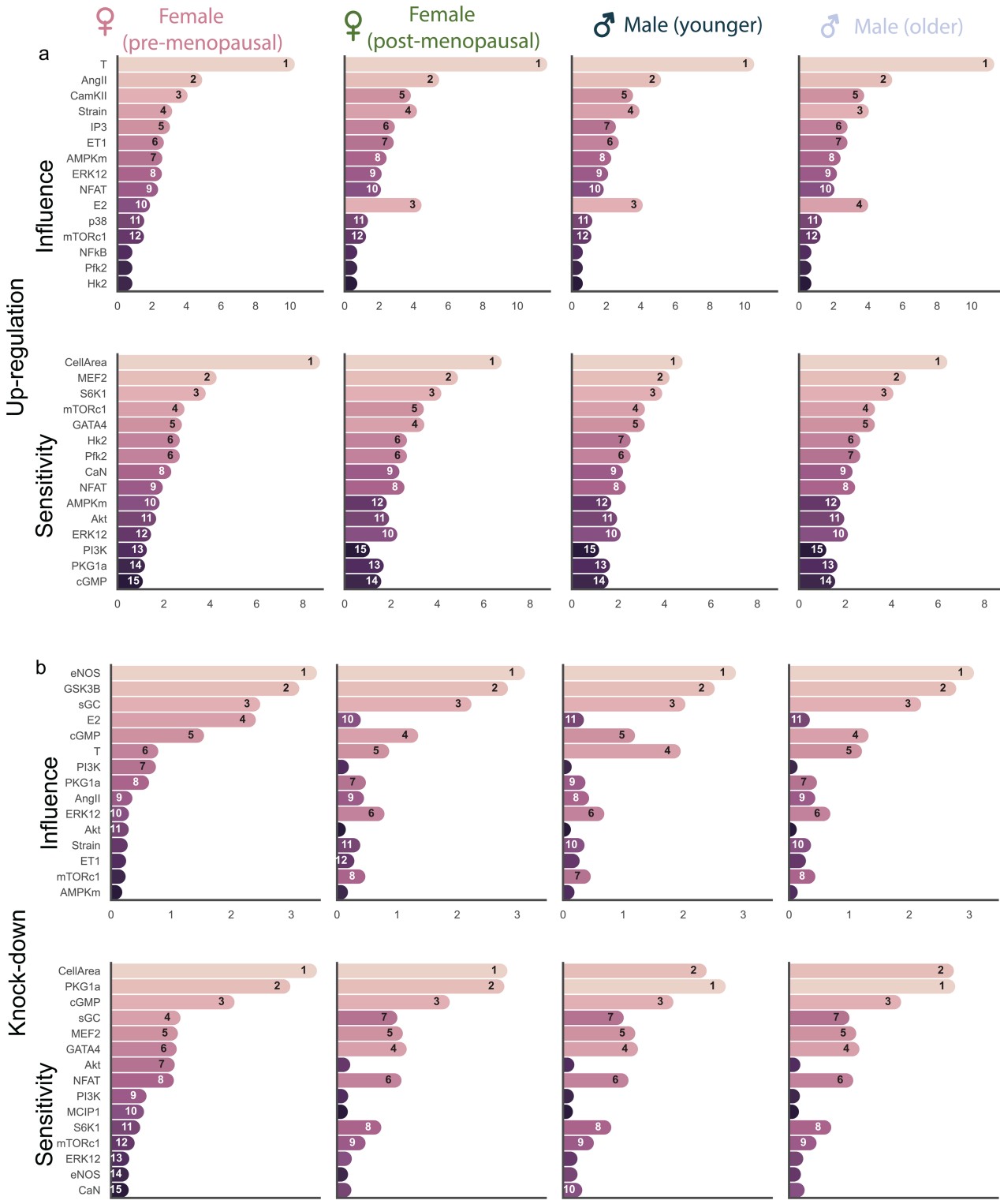

**Fig 6. Model response to up-regulations and knock-downs: Analysis of the impact of sex hormones on hypertrophic signaling in pre- and post-menopausal females, and younger and older males identifies top 15 influential nodes that drives remodeling, and the 15 nodes most sensitive to these (a) Up-regulation and (b) Knock-down perturbations.** The nodes are sorted based on their rank in the pre-menopausal female model, with their rank in each model indicated inside the bars.

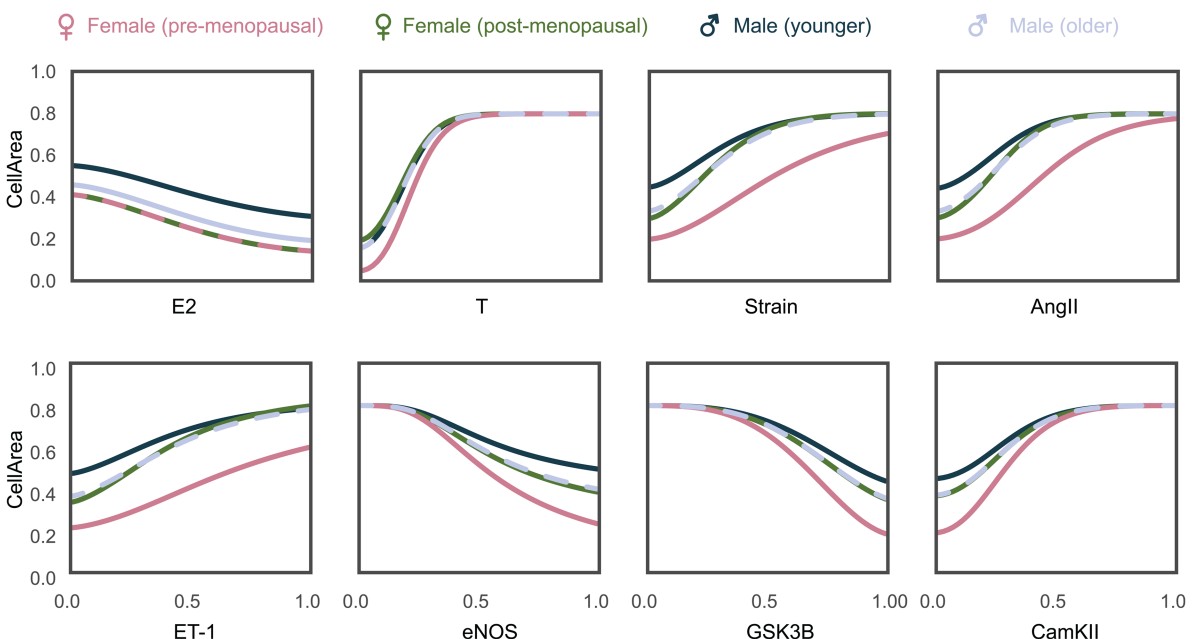

**Fig 7**. **Continuous node behavior of input nodes (E2, T, Strain, AngII, ET-1) and some intermediary nodes (eNOS, GSK3β, and CamKII) on the output, CellArea for pre- and post-menopausal females, younger and older males.**

## Discussion

We developed a logic-based systems biology model to investigate how sex hormones (E2 and T) interact with established hypertrophic stimuli (Strain, AngII and ET-1) to drive cardiomyocyte hypertrophy. The model was qualitatively validated against literature data used to build the model. The model was also quantitatively validated against three independent studies, successfully reproducing known sex hormone-dependent behaviors. Analysis of the model under pre-menopausal female, post-menopausal female, younger male and older male hormone concentrations revealed key regulatory nodes that respond differently to varying hormone levels. Our sensitivity and influence analyses suggest that differences in T may be more important than E2 in explaining age-like and sex-specific differences in susceptibility to hypertrophic car-diomyopathy.

## Comparison with prior work

Existing computational models have successfully simulated organ-scale cardiac remodeling using mechanics-driven growth laws [59–65]. However, these models cannot explain underlying mechanisms on the sub-cellular scale. Moreover, these mechanics-driven models do not account for the impact of medications on cardiac remodeling, a significant limi-tation considering that most heart disease patients are under pharmacological treatment. Logic-based systems biology models have been able to fill this gap successfully by exploring cell-level remodeling mechanisms in cardiomyocytes via intracellular signaling pathways [14,17,18]. The most recent model by Frank et al. [18] analyses cardiomyocyte hyper-trophic signaling using 106 species and 192 reactions. However, none of these models considered the effect of sex hor-mones. Due to limited evidence of sex hormone influence on many pathways in existing large-scale network models, we developed a simplified model focusing exclusively on sex hormone-responsive pathways. For example, known hyper-trophic stimuli such as ANP, BNP, JNK, and JAK/STAT etc. lacked studies demonstrating sex hormone effects on these

pathways. Note that limited experimental evidence does not preclude sex hormone effects on these interconnected pathways, especially because many of these pathways are interconnected downstream. Our aim was to focus solely on hypertrophy, thus excluding contractility from the current model. However, it is important to note that hypertrophy and contractility are closely intertwined, with overlapping species and pathways. As a result, completely decoupling hypertrophy from contractility is challenging, and some overlap may still exist in the model.

Recent studies have applied logic-based modeling to investigate the role of E2 in cardiac fibroblast drug responses [21] and to analyze E2 and progesterone effects on cardiomyocyte hypertrophy during and after pregnancy [22,66]. While our model does not include progesterone, it adds additional stimuli and pathways not present in existing models, especially those pertaining to T and its downstream pathways, as well as additional cellular energetics nodes such as AMPK, etc. Our model shares key nodes and pathways with these models, including E2, AngII, mechanical strain, CaN, ERK1/2, and Akt.

To the best of our knowledge, the study presented here is the first network model to incorporate T, and mechanistically investigate the role of both E2 and T on hypertrophy together within the same network. The sex specific combination of E2 and T has been shown to have significant impacts on cardiovascular health in both males and females, with imbalances associated with various cardiovascular diseases and conditions [67–73]. While studies examining E2 and T simultaneously are lacking, understanding their combined effects is crucial given that both hormones are present in varying levels across sexes and ages. Further research is needed to fully characterize these hormone interactions within hypertrophy networks.

## Physiological implications

The influence and sensitivity analysis revealed interesting influences of E2 and T on cardiomyocyte hypertrophic signaling. In the up-regulation analysis (Fig 6a), the influence of T and AngII are highest for all four models. However, their influence magnitudes are slightly higher in the post-menopausal female and male models. T is the most influential node in all models, but it has slightly higher level of influence in post-menopausal females and older males. AngII is seen to be the second most influential node, with a higher influence in the male and post-menopausal female models, which makes a case for sex- and age-specific dosing.

The increased CamKII influence across all models indicates an increase in calcium signaling activity [74,75]. In future models that incorporate contractility, we can further explore this enhanced calcium signaling activity. Such investigation might provide insights into why healthy females typically exhibit lower contractility than healthy males, while in the presence of pre-existing hypertrophy, the contractility between sexes becomes comparable [76]. We see this reflected in the pre-menopausal model where CamkII's higher influence during up-regulaton hints at being much lower at baseline than the post-menopausal and male models.

In the knock-down analysis (Fig 6b), eNOS and GSK3$\beta$ emerge as the most influential nodes across all models. However, the magnitude of eNOS and GSK3$\beta$ influence is slightly higher in pre-menopausal females. eNOS is a vasodilator that regulates heart rate [77], and ameliorates progression of left ventricular hypertrophy [78]. GSK3$\beta$ localizes to the cardiac Z-disc to maintain length-dependent activation in cardiomyocytes [79]. Both eNOS and GSK3$\beta$ are known to have anti-hypertrophic behavior, which is evident in Fig 7. Notably, the post-menopausal female curve is closer to both the male curves rather than the pre-menopausal female curve.

Based on the combined results of the influence/sensitivity analysis and case studies, we expect that T will have a larger effect on cardiomyocyte hypertrophy than other input species. In the model this implies that a certain change in T will cause a bigger change in CellArea than the same change in other inputs. Biologically this implies that physiological changes in T will have a larger impact on cardiac hypertrophy than the other inputs, as seen in puberty where T increases in boys and E2 increases in girls, yet only the heart size of boys deviates from the linear growth curve observed up to that point.

We consider a decrease in E2 in females and also explore age-related decreases in T in males, although this change is much smaller (around 10%) compared to the decrease of E2 in females (around 60%) from 45 to 65 years of age while transitioning to menopause [12]. More detailed age-related changes in signaling pathways should supplement this model's predictions in the future.

## Beyond sex hormones

Our model mainly focused on hormone-driven signaling pathways, ignoring potential genetic differences between male and female cells. Prior studies have used differential gene expression analysis to identify protein kinase A (PKA) [80] and Peroxisome proliferator-activated receptor gamma (PPAR-γ) [81] as important clusters to understand age and sex differences in cardiac hypertrophy. While these studies hypothesize that E2 alone is unlikely to explain the differences in female gene expression across the course of life, they do not delve into T, or mechanistic analysis of the cross-talk between E2 and T. It is also important to note that sex differences in cardiac remodeling extend beyond gonadal hormones. Factors such as sex chromosomes, epigenetics, and autosomal gene regulation variations [2–4] require further research for a more comprehensive understanding of the role of chromosomes and other genetic factors. Interestingly, all the nodes in our current sex-driven hypertrophy model are encoded by autosomal genes. This means these nodes are present in both males and females, making the model applicable to both sexes.

## Limitations

Our model has several limitations. First, the predictions of our model are only as complete as the experimental data that is available. Unfortunately, our search was limited by the fact that many studies on cardiac hypertrophy do not report sex, or sex hormones in their results. We anticipate that this could be improved in the future, since major funding agencies such as the National Institutes of Health (USA), European Research Commission and Canadian Institutes of Health Research have only mandated considering sex as a variable since the 2010s. Thus, there might be additional pathways and nodes downstream of E2 and T that affect hypertrophy but are not included in the current model due to lack of experimental evidence. This omission could have led to over- or under-representation of either E2 or T in our model, which might have contributed to under- and over-estimation in model predictions in (Fig 3). Interestingly, in our model both E2 and T are linked to a similar number of downstream nodes, meaning the greater influence of T compared to E2 is not due to a connectivity bias. Other sex hormones such as progesterone, follicle-stimulating hormone, and luteinizing hormone were not considered due to lack of experimental studies on their impact on cardiomyocyte hypertrophy, except in the context of pregnancy.

The second limitation is the model parameterization. Our model contains 47 Hill-type reactions governed by 3 kinetic parameters, so we have a total of 141 parameters. Identifying each Hill parameter would require quantitative data for each single reaction. While there have been some studies that determined the kinetic parameters for a single reaction pathway, such as ET-1's effect on CaN in cardiac hypertrophy [82], rate kinetics of most of the individual reactions were unknown. Instead, we identified most of the reactions in our model from qualitative data such as from Western Blots. Thus, we assumed the same Hill parameters for all reactions, which is consistent with others who have successfully used this modeling approach [14,17,18,22,52,83,84]. In fact, this approach can be considered a strength of logic-based models, as they enabled the prediction of quantitative changes in cell area (Fig 3a) and (Fig 3b) despite being constructed solely from qualitative data.

Third, our current model does not include an organ-scale component to simulate changes in strain due to changes in hemodynamics. Since we do not have an organ-scale model to simulate the change in strain due to these changes in hemodynamics, we assumed its increase to be similar to Yoshida et al. [22]. This is likely the cause of the over- and under-estimations of strain-induced hypertrophy for the in vivo studies in our quantitative validation (Fig 3). In these studies, hypertrophy was induced by pressure overload (through transverse aortic constriction) or hypertension. Recent work

from Estrada et al. [83] coupled the cardiomyocyte hypertrophic signaling model from Ryall et al. and Frank et al. [17,18] to a finite element model of the left ventricle to address this limitation.

Fourth, the CellArea output in our model is very sensitive to input changes. We believe that this is a consequence of having one single composite output for the model. So, more nuanced signaling pathways pertaining to synthesis and degradation of various biomolecules could lead to more robust behavior. It could also be due to a lack of feedback loops to stabilize the output. For example, in the multiscale model formulation by Estrada et al. [83] discussed above, an increase in CellArea leads to a reduction in Strain because of tissue thickening, thus stabilizing cardiomyocyte hypertrophy. Our calibrated model inputs are situated on the steeper part of the Hill curves, causing any perturbation to have a relatively large downstream effect, thereby increasing the sensitivity of our output. In our model, CellArea is a simplified representation of more nuanced changes in biomolecules, such as actin and myosin. However, since their ratios would have implications for cellular behavior and function such as contractility, we did not include them in our current model of hypertrophy. For Strain, a composite input, we could not find studies on the impact of sex hormones on hypertrophy-related integrins and ion channels that together sense and translate changes in the cell's mechanical environment.

Finally, female hormone concentrations are cyclical (e.g. during the menstrual cycle), but are kept constant in the analysis. Future models could benefit from considering the temporal variations in hormone concentrations , such as incorporating the menstrual cycle, or delineating the different stages of menopause, or changing medication doses [85].

## Future considerations

In this study, we developed a model to study the impact of sex hormones on cardiomyocyte hypertrophic signaling. A similar approach, likely involving many of the same pathways, could be applied in the future to incorporate sex hormone-driven changes in other key aspects of cardiac function, such as contractility. There is also a need to couple the cell-level model to an organ-scale model, as discussed in the Limitations, originally proposed by Estrada et al. [83] and since adopted by others to model cardiac and vascular remodeling [22,66,84]. This would allow modeling of closed-loop feedback interactions between changes in cell size, cardiac mass, and strain, enabling the identification of multi-scale mechanisms and improving patient-specific predictions. This coupled model will enable assessing the influence of medication patients are on in a sex-specific manner, and might also improve the current network model's simulation of in vivo data.

Integrating transcriptomic information into our mode, such as by adjusting reaction weights, in the future could further improve the precision of future treatments for cardiac hypertrophy. Sex-specific transcriptomic signatures observed at the single-cell level lead to differential gene expression patterns, thereby influencing cellular hypertrophic response to the hormonal inputs present in this network model. In the future, we see our model being used to answer questions such as understanding how hormone replacement therapy (HRT) impacts cardiac remodeling in post-menopausal females, and why E2 supplementation to pre-menopausal levels does not improve health outcomes [86]. We would also like to use our model to understand the unknown outcomes of gender affirming hormone therapy (GAHT) on the heart [87].

In conclusion, this study provides a critical tool towards understanding the sex-specific regulation of cardiac hypertrophy, laying the groundwork for future research into multi-scale modeling and personalized cardiac health interventions. As heart failure patients exhibit both mechanical and sex-specific hormonal changes, and receive drugs that alter sex-specific pathways, our model has the potential to lead to advancements in heart failure therapies for both women and men.

## Methods

### Logic-based normalized Hill equations

Model reactions were modeled using the logic-based normalized Hill approach from Kraeutler et al. [14]. In this framework, every model node $X$ takes a value between $y_{min} = 0$ and $y_{max} = 1$, representing complete deactivation or activation, respectively. The two main processes in the network model are activation and inhibition. Sigmoidal activation or inhibition

by a single upstream model node $X \in [0, 1]$ is modeled by normalized Hill functions:

$$f_{act}(X) = W\frac{BX^n}{K^n + X^n}, \quad f_{inh}(X) = W - f_{act}(X), \tag{1}$$

with $n$ the Hill coefficient, $W \in [0, 1]$ a reaction weight, and the constants $B$ and $K$ defined as

$$B = \frac{EC_{50}^n - 1}{2EC_{50}^n - 1}, \quad K = (B - 1)^{1/n}, \tag{2}$$

where $EC_{50}$ is the value of node $X$ at which a half-maximal activation occurs, i.e. $f_{act}(EC_{50}) = f_{inh}(EC_{50}) = 0.5$. $K$ and $B$ also enforce the activation function to become

$$f_{act}(0) = 0, \quad f_{act}(1) = 1. \tag{3}$$

Conversely, the inhibition function becomes

$$f_{inh}(0) = 1, \quad f_{inh}(1) = 0. \tag{4}$$

Following common assumptions in logic-based network modeling, we set uniform Hill parameters across all reactions, setting $EC_{50} = 0.5$ and $n = 1.4$ following Ryall et al. [17].

For the generic composite nodes that aren't biomolecules such as Strain which act on several molecules, a value of 0 means Strain has no effect on downstream reactions, while a value of 1 means a maximum effect. For the generic output node, CellArea, a value of 0.5 represents the steady-state baseline, with 0 representing the lowest possible cell size and 1 representing the maximum possible cell size.

Multivariable activation or inhibition was modeled using AND and OR operators. For example, for activation by nodes $X$ AND $Y$ and $X$ OR $Y$:

$$X \wedge Y = f_{act}(X)f_{act}(Y), \tag{5}$$
$$X \vee Y = f_{act}(X) + f_{act}(Y) - f_{act}(X)f_{act}(Y). \tag{6}$$

These operators can be used recursively to create more complex relationships. In Fig 1 the AND gates are shown by '&' and OR gates exist when more than 1 node influences a downstream node.

The activation and inhibition functions together with logic operators were used to model the reactions in the network model as a system of ordinary differential equations (ODEs). First, consider the simple activation of $IP3$ by $T$. Using the normalized Hill functions, the change in activation of $IP3$ is defined as:

$$\frac{dIP3}{dt} = \frac{1}{\tau_{IP3}}\left[f_{act}(T)y_{max,IP3} - IP3\right], \tag{7}$$

with $\tau_{IP3}$ the species time constant and $y_{max,IP3}$ its maximum activation. By default, $\tau$ is set to 0.1 h for all species in our model except for $\tau_{CellArea} = 300$ h, and $y_{max}$ is set to 1, similar to [22]. Now consider the more complex regulation of NFAT, where we model both its synergistic activation by ERK1/2 and CaN and its inhibition by GSK3β using AND/OR logic gates:

$$\frac{dNFAT}{dt} = \frac{1}{\tau_{NFAT}}\left[\left(\left(f_{act}(ERK1/2) \wedge f_{act}(CaN)\right) \vee f_{inh}(GSK3\beta)\right)y_{max,NFAT} - NFAT\right]. \tag{8}$$

The system of ODEs for our model was generated by the open-source tool Netflux [25].

## Model calibration

We identified values for the weights of the input species (E2, T, AngII, ET-1 and Strain) that resulted in a CellArea of 0.5 at baseline physiological conditions. Since like all other species, CellArea $\in [0, 1]$, this allows for equal amounts of pro-hypertrophic and anti-hypertrophic behavior. Multiple parameter combinations could achieve this equilibrium, as seen in Fig 4, so we performed the calibration by constraining all inputs to a single value, which converged to $w_{E2} = w_T = w_{AngII} = w_{ET-1} = w_{Strain} = 0.15$. Younger and older male and pre- and post-menopausal input weights used in Figs 6 and 7 were calibrated using known changes in heart size across puberty and menopause. For younger males, we set $w_T = 0.21$ to achieve a CellArea of 0.65, reflecting the post-pubertal 1.3-fold increase in male left ventricular wall mass relative to females [8]. For older males, we set $w_T = 0.17$ to reflect a 20% reduction and achieve a CellArea of 0.53. Similarly, for post-menopausal females we set $w_{E2} = 0.01$ to achieve a CellArea of 0.54, reflecting the post-menopausal 1.08-fold increase in female left ventricular wall mass relative to pre-menopause [88]. Since there was no clear developmental stage to calibrate the increase of E2 in pre-menopausal women, we chose to set $w_{E2} = 0.5$, which is in the center of its range, leaving space for higher degrees of up-regulation, such as during pregnancy and the ovulation phase of the menstrual cycle and pregnancy. All other input weights ($w_{AngII}, w_{ET-1}, w_{Strain}$) remained at baseline values.

## Model validation

We validated the model against qualitative and quantitative data. For the qualitative validation (Fig 2), we compared model predictions against individual perturbations and their modulations with E2 or T reported in the literature. Given the qualitative nature of available data, we used standardized node upregulation of $\Delta y = 0.3$. Model agreement with the qualitative literature was determined by comparing the decrease ($\Delta y < -0.05$), increase ($\Delta y > 0.05$), or no change ($-0.05 \leq \Delta y \leq 0.05$) of node activity with literature data. The upregulation and threshold values were based on Irons et al. [52].

For the quantitative validation, we compared model predictions against one in vitro and two in vivo studies. We simulated serum increases of AngII and ET-1 in the in vitro study [53] by setting their reaction weights to the near-maximum value of $w_{AngII} = w_{ET-1} = 0.9$, mimicking their high serum concentration. Pressure overload by aortic banding is known to result in an increase in strain, so we simulated the rat pressure overload study by Donaldson et al. [54] by setting $w_{Strain} = 0.6$, similar to Yoshida et al. [22]. As AngII is known to increase in response to aortic banding, we increased $w_{AngII}$ by a factor two of its baseline value. The in vivo study of spontaneously hypertensive rats by [55] was simulated using an additional increase of $w_{AngII}$ by a factor two of its baseline value. As Strain is known to increase in response to spontaneous hypertension, we set $w_{Strain} = 0.6$. E2 and T perturbations were simulated by setting $w_{E2} = 0.5$ and $w_T = 0.21$ for the in vitro studies. These values enforce approximate half-maximum increases, since we only expect maximum E2 concentrations in situations such as pregnancy [22]. We adjusted the baseline $w_{E2} = 0.5$ and an E2 increase of $\Delta y_{E2} = 0.3$ for the in vivo female rats without OVX (ovariectomy) in Donaldson et al. [54] and baseline $w_{E2} = 0.01$ and an E2 increase of $\Delta y_{E2} = 0.3$ for the in vivo female rats with OVX in [55]. All predictions were normalized with respect to the calibrated baseline model. To calculate the standard deviation for the normalized change, we used the propagation of error method from Ku [89].

## Influence and sensitivity analysis

To identify influence and sensitive mechanisms across our model, we simulated a series of knock-down and up-regulation perturbations, following prior approaches in logic-based model analysis [21,58,90]. Influence $I_i$ of node $i$ was defined as the sum of absolute changes in activity of all other species $j$ when species $i$ is perturbed:

$$I_i = \sum_{j \neq i} |\Delta y_{ij}| \quad \text{where} \quad \Delta y_{ij} = y_j(X_{i,\text{perturb}}) - y_j(X_{i,0}). \tag{9}$$

Conversely, sensitivity $S_i$ of species $i$ was defined as the sum of absolute change in activity of the species across knockdown/up-regulation of all other species $j$:

$$S_i = \sum_{i \neq j} |\Delta y_{ij}| \quad \text{where} \quad \Delta y_{ij} = y_i(X_{j,\text{perturb}}) - y_i(X_{j,0}). \tag{10}$$

Up-regulation was simulated by setting $X_{\text{perturb}} = 0.8$ and knockdown by setting $X_{\text{perturb}} = 0.1$ [90]. The time constant of perturbed nodes was set to $\tau \to \infty$ to enforce the perturbation. This analysis was repeated for the pre- and post-menopausal female, and, younger and older male models to determine the impact of sex hormones on mechanisms of cardiomyocyte hypertrophy. To assess each input parameter's contribution to the variance of the model output, we performed a global first-order Sobol sensitivity analysis using the Saltelli method in the SALib python package. Note that Sobol's method, like most global sensitivity analyses, assumes that inputs are independent. In our model as several downstream nodes are dependent on each other through interconnected signaling pathways, we only include the most upstream nodes as inputs in the Sobol analysis.

## Code and data availability

All model code and data used to generate the results in this study will be available on our lab's GitHub page (http://www.github.com/BEATLabUCI) upon publication of this manuscript. This repository includes: Python code developed to run and analyze the model; annotated Jupyter notebooks that provide guidance on using the code and reproducing the results and figures; Excel file containing all species and reactions, and their parameters and sources.

## Acknowledgments

We thank our lab members for their helpful discussions and feedback.

## Author contributions

**Conceptualization:** Adhithi Lakshmikanthan, Pim Oomen.

**Data curation:** Adhithi Lakshmikanthan, Minnie Kay.

**Formal analysis:** Adhithi Lakshmikanthan, Pim Oomen.

**Funding acquisition:** Pim Oomen.

**Investigation:** Adhithi Lakshmikanthan, Minnie Kay, Pim Oomen.

**Methodology:** Adhithi Lakshmikanthan, Pim Oomen.

**Project administration:** Pim Oomen.

**Resources:** Pim Oomen.

**Software:** Adhithi Lakshmikanthan, Pim Oomen.

**Supervision:** Pim Oomen.

**Validation:** Adhithi Lakshmikanthan, Pim Oomen.

**Visualization:** Adhithi Lakshmikanthan, Pim Oomen.

**Writing – original draft:** Adhithi Lakshmikanthan, Minnie Kay, Pim Oomen.

**Writing – review & editing:** Adhithi Lakshmikanthan, Minnie Kay, Pim Oomen.

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
