## [Decision Letter · Decision Letter 0]

9 Jul 2025

PCOMPBIOL-D-25-00341

Modeling the Interplay of Sex Hormones in Cardiac Hypertrophic Signaling

PLOS Computational Biology

Dear Dr. Oomen,

Thank you for submitting your manuscript to PLOS Computational Biology. After careful consideration, we feel that it has merit but does not fully meet PLOS Computational Biology's publication criteria as it currently stands. Therefore, we invite you to submit a revised version of the manuscript that addresses the points raised during the review process.

Please submit your revised manuscript within 60 days Sep 08 2025 11:59PM. If you will need more time than this to complete your revisions, please reply to this message or contact the journal office at ploscompbiol@plos.org. Please include the following items when submitting your revised manuscript:

We look forward to receiving your revised manuscript.

Kind regards,

Juilee Thakar

Academic Editor

PLOS Computational Biology

Pedro Mendes

Section Editor

PLOS Computational Biology

**Additional Editor Comments :**

Dear Dr. Oomen,

Thank you for your patience. Although, there is a lot of enthusiasm about your manuscript, two reviewers have provided critical feedback and I would appreciate if you can respond to those in detail.

Best,

Juilee Thakar

**Journal Requirements:**

3) Please amend your detailed Financial Disclosure statement. This is published with the article. It must therefore be completed in full sentences and contain the exact wording you wish to be published.

**Reviewers' comments:**

Reviewer's Responses to Questions

**Comments to the Authors:**

**Please note that three reviews are uploaded as attachments.**

Reviewer #1: The review is uploaded as an attachment.

Reviewer #2: Attached

Reviewer #3: THE REVIEW IS UPLOADED AS A PDF ATTACHMENT

Reviewer #4: Sex has been increasingly recognized as a key biological variable influencing physiological behaviors and plays a critical role in diseases, including cardiovascular conditions such as hypertrophy. While many of these effects are mediated by sex hormones, their precise mechanistic impact on cardiac hypertrophy development remains understudied.

In this manuscript, Dr. Lakshmikanthan and colleagues present a cardiomyocyte model that elucidates the effects of sex hormones on cardiomyocyte hypertrophy, quantified by myocyte area. The model's predictions are rigorously validated against prior experimental observations. Their findings demonstrate that testosterone exerts a greater impact than estradiol on cardiac hypertrophy. This study offers novel mechanistic insights and a computational tool to enhance our understanding of sex hormone effects on hypertrophy. The model-building and validation approaches align with established methodologies in the field.

While the reviewer acknowledges the model's novelty and value, the manuscript could be strengthened by addressing the following areas:

1. Lack of formal sensitivity analysis (e.g., Sobol’s indices) to assess uncertainties in the newly added network nodes. Such an analysis would clarify the most influential nodes and pathways mediating sex hormone effects in male and female hearts, a standard practice in prior studies. This approach could uncover key nodal signals and clustered pathways critical to hypertrophy, deepening mechanistic insights.

2. Interpretation of hormone impact comparisons—The model's major novel finding is that testosterone has a greater effect than estradiol on hypertrophy. However, simulations rely on normalized upregulations/knockdowns rather than exact hormone concentrations or nodal expression levels. Thus, the physiological relevance of these comparisons remains unclear.

3. Aging considerations—Aging (e.g., pre- vs. post-menopausal females, males) is modeled solely by adjusting sex hormone levels. Yet, aging involves complex changes in multiple nodal signals within the network. The extent to which this simplification reflects real-world cardiac aging is uncertain. Given that cardiac hypertrophy is age-associated—and aging reduces testosterone in males—the model’s ability to explain this observation warrants further exploration.

4. Potential therapeutic targets—As noted, the manuscript could be enhanced by investigating druggable targets to mitigate hypertrophy in aged males and females, offering additional translational relevance.

**Have the authors made all data and (if applicable) computational code underlying the findings in their manuscript fully available?**

Reviewer #1: Yes

Reviewer #2: **No: **The manuscript indicates the code is available, but I could not find it in the provided link

Reviewer #3: **No: **While the submission provides the lab's github page it does not point to a particular directory for the project. Nor does any of the available repositories seem to correspond to this manuscript.

Reviewer #4: **No: **

PLOS authors have the option to publish the peer review history of their article (what does this mean?). If published, this will include your full peer review and any attached files.

Reviewer #1: No

Reviewer #2: No

Reviewer #3: **Yes: **Luis A. Álvarez-García

Reviewer #4: No

**Figure resubmission:**
---

## [Decision Letter · Decision Letter 1]

29 Sep 2025

PCOMPBIOL-D-25-00341R1

Modeling the Interplay of Sex Hormones in Cardiac Hypertrophic Signaling

PLOS Computational Biology

Dear Dr. Oomen,

Thank you for submitting your manuscript to PLOS Computational Biology. After careful consideration, we feel that it has merit but does not fully meet PLOS Computational Biology's publication criteria as it currently stands. Therefore, we invite you to submit a revised version of the manuscript that addresses the points raised during the review process.

Please submit your revised manuscript within 30 days Nov 29 2025 11:59PM. If you will need more time than this to complete your revisions, please reply to this message or contact the journal office at ploscompbiol@plos.org. Please include the following items when submitting your revised manuscript:

We look forward to receiving your revised manuscript.

Kind regards,

Juilee Thakar

Academic Editor

PLOS Computational Biology

Pedro Mendes

Section Editor

PLOS Computational Biology

**Additional Editor Comments:**

Thank you for responding to most of the comments. Reviewer 4 has raised important concerns. Could you please respond to one of the outstanding concerns?

Best,

Juilee

**Journal Requirements:**

**Reviewers' comments:**

Reviewer's Responses to Questions

**Comments to the Authors:**

Reviewer #2: The authors have addressed my previous comments in the new version of the manuscript.

Reviewer #4: The reviewer would like to thank the authors for carefully considering the previous comments from the reviewer. However, as most of the previous comments could be addressed by additional set of simulations but it appears that the responses are mainly descriptive without new data. I would like to request more responsive revision for the following comments.

* Aging and sex hormone levels - "We chose to not explore age-related decreases in T in males since this change is much smaller (~10%)". Additional simulations with 10%-20% reduction with aging would be essential to help consolidate the claimed conclusions. This is important because the sensitivity analysis does indicate a strong influence of T on the investigated components related to hypertrophy.

* Potential therapeutic targets - the authors describes compounds "valsartan/ sacubitril" in the responses. It would valuable without requiring excessive effort to simulate the effects of these drugs using the model.

* Since the aging model appears to lack detailed descriptions of aging-related changes in the cardiomyocytes, it is recommended to tone down the description of aging model in the manuscript, and replacement by 'aging-like' would be more faithful.

**Have the authors made all data and (if applicable) computational code underlying the findings in their manuscript fully available?**

Reviewer #2: Yes

Reviewer #4: Yes

PLOS authors have the option to publish the peer review history of their article (what does this mean?). If published, this will include your full peer review and any attached files.

Reviewer #2: No

Reviewer #4: No

**Figure resubmission:**
---

## [Editor Report · Decision Letter 2]

1 Dec 2025

Dear Assistant Professor Oomen,

We are pleased to inform you that your manuscript 'Modeling the Interplay of Sex Hormones in Cardiac Hypertrophic Signaling' has been provisionally accepted for publication in PLOS Computational Biology.

Best regards,

Juilee Thakar, Ph.D.

Academic Editor

PLOS Computational Biology

Pedro Mendes

Section Editor

PLOS Computational Biology

---

## [Editor Report · Acceptance letter]

PCOMPBIOL-D-25-00341R2

Modeling the Interplay of Sex Hormones in Cardiac Hypertrophic Signaling

Dear Dr Oomen,

I am pleased to inform you that your manuscript has been formally accepted for publication in PLOS Computational Biology. Your manuscript is now with our production department and you will be notified of the publication date in due course.

With kind regards,

Olena Szabo
